# Intraoperative Augmented Reality in Microsurgery for Intracranial Arteriovenous Malformation: A Case Report and Literature Review

**DOI:** 10.3390/brainsci13040653

**Published:** 2023-04-13

**Authors:** Chi-Ruei Li, Chiung-Chyi Shen, Meng-Yin Yang, Yuang-Seng Tsuei, Chung-Hsin Lee

**Affiliations:** Department of Neurosurgery, Neurological Institute, Taichung Veterans General Hospital, Taichung 40705, Taiwan; fantastic1694@gmail.com (C.-R.L.); ccshen61093@gmail.com (C.-C.S.); yangmy04@gmail.com (M.-Y.Y.); astrocytoma2001@yahoo.com.tw (Y.-S.T.)

**Keywords:** augmented reality (AR), arteriovenous malformation (AVM), digital subtraction angiography (DSA), hybrid operating room, microsurgery

## Abstract

Background: Intracranial arteriovenous malformations (AVMs) are lesions containing complex vessels with a lack of buffering capillary architecture which might result in hemorrhagic cerebrovascular accidents (CVAs). Intraoperative navigation can improve resection rates and functional preservation in patients with lesions in eloquent areas, but current systems have limitations that can distract the operator. Augmented Reality (AR) surgical technology can reduce these distractions and provide real-time information regarding vascular morphology and location. Methods: In this case report, an adult patient was admitted to the emergency department after a fall, and diagnostic imaging revealed a Spetzler–Martin grade I AVM in the right parietal region with evidence of rupture. The patient underwent a stereotactic microsurgical resection with assistance from augmented reality technology, which allowed for a hologram of the angioarchitecture to be projected onto the cortical surface, aiding in the recognition of the angiographic anatomy during surgery. Results: The patient’s postoperative recovery went smoothly. At 6-month follow-up, the patient had remained in stable condition, experiencing complete relief from his previous symptoms. The follow-up examination also revealed complete obliteration of the AVMs without any remaining pathological vascular structure. Conclusions: AR-assisted microsurgery makes both the dissection and resection steps safer and more delicate. As several innovations are occurring in AR technology today, it is likely that this novel technique will be increasingly adopted in both surgical applications and education. Although certain limitations exist, this technique may still become more efficient and precise as this novel technology its continues to develop further.

## 1. Introduction

Intracranial arteriovenous malformations (AVMs) are lesions containing complex vessels with a lack of buffering capillary architecture, which causes direct arterial shunting to drainage veins [1]. This pathological shunting network can result in hemorrhagic cerebrovascular accidents (CVAs), having an estimated incidence rate of 2–4% [2,3,4]. According to the Spetzler–Martin grading scale [5], the literature has proposed the use of microsurgery as a feasible option for patients with Grade I or Grade II lesions which possess 2% morbidity and 0.3% mortality rates, respectively [5,6,7].

For microsurgical resection of AVMs, intraoperative navigation can provide detailed information for angioarchitecture, and through its assistance, the total resection rate can be significantly elevated during microsurgical treatments for patients [8]. Particularly in cases where lesions are located in an eloquent area, i.e., the sensory, motor, language or visual cortex, navigation-assisted microsurgical resection can achieve promising results in both total resection rate and functional preservation [9].

Nevertheless, these navigational systems are limited due to shortcomings that shift the attention of the operator. For example, neurosurgeons may get distracted due to the need to look away from the surgical field seen under the microscope in order to focus attention on an external monitor. To address any limitations these systems may cause, Augmented Reality (AR) surgical technology was designed to reduce distracting factors, while also providing surgeons with real-time feedback regarding vascular morphology and location [10,11,12,13]. In this case report, we present a patient who has been diagnosed with intracranial AVM and subsequently received AR-assisted microsurgery for lesion resection.

## 2. Illustrative Case

An adult patient was brought to our emergency department due to a falling accident. According to a witness, this patient suffered from a sudden change in consciousness, followed by a 3 m height fall while working. Upon arrival at our emergency department, his Glasgow Coma Scale (GCS) was E4V5M6, without any signs of specific neurological deficit. A review of his medical history showed that he had an incidental right parietal lesion with hemorrhagic vestige (Figure 1A) which was revealed through computed tomography(CT) 6 months earlier during a hospital visit for intermittent headaches. No weakness in extremities or neurological deficit was noted during this period.

Diagnostic workup involving magnetic resonance imaging (MRI) and magnetic resonance angiography (MRA) revealed a lesion with a tangle of vessels (Figure 1B) presenting itself with a signal void appearance in the right parietal region on a T2-weighted (Figure 1C) series. In addition, the Spetzler–Martin grade I AVM (size < 3 cm, superficial drainage, non-eloquent cortex) revealed evidence of a rupture with hemorrhage. The main arterial supply was comprised in the right terminal branch of the posterior cerebral artery and the parietal branch of the middle cerebral artery, with superficial drainage into cortical vein superiorly.

## 3. Operation

The patient was prepped to undergo a stereotactic microsurgical resection. A previously obtained brain MRI with contrast was integrated into the stereotactic neuronavigation system (BrainLab Curve) (BrainLab AG, Munich, Germany). Prior to surgery, we schemed the feeding arteries and AVM nidus in different colors on a 3D reconstructive model and merged them with MRI images.

The patient was positioned supine on the operating table with the head turned leftward before being fixed with a skull clamp. The reference cluster for navigation recognition was attached to the head frame for registration. Following the registration procedures, the other stereotactic reference was affixed to the microscope (KINEVO 900, Carl Zeiss AG, Oberkochen, Germany).

The first stage of the operation for the patient’s AVM resection was confirmative digital subtraction angiography (DSA) (Figure 1D) followed by right parieto-occipital craniotomy. After performing a dura opening, surgery proceeded to the microsurgical stage performed under a microscope (Figure 2A). With assistance from the AR technique, the feeding arteries from the posterior cerebral artery and the parietal branch of the middle cerebral artery were labeled with red and yellow color, respectively, and then projected onto the cortex. Furthermore, the AVM nidus was coded with blue color for identification (Figure 2B). In the next dissection and resection stage, we adjusted the focal length and hologram setting to clarify the surgical field. After confirmation of the spatial relationship in the angiographic complex, both coagulation and clipping were performed to obliterate the feeding artery (Figure 2C). After the clipping and coagulation of the first feeding artery from the middle cerebral artery, we proceeded to obliterate another feeding artery from the posterior cerebral artery (Figure 2D). Eventually, the AVM nidus got resected after both the feeding arteries were obliterated.

## 4. Postoperative Course

The patient’s postoperative recovery went smoothly, and he was subsequently transferred to the intensive care unit for further care and monitoring. Extubation was performed the day after surgery, with the patient’s clinical condition remaining stable and no newly onset neurological deficit being noted. At 6-month follow-up, the patient had remained in stable condition, experiencing complete relief from his previous symptoms. The follow-up examination also revealed complete obliteration of the AVMs without any remaining pathological vascular structure.

## 5. Discussion

In our case, the surgical set up resembled the microsurgical resections performed by Scherschinski et al. [13] and Cabrilo et al. [14] who had integrated preoperative radiological data with the stereotactic navigation process. The virtual anatomical hologram projecting onto the cortical surface helps the operator identify complex angioarchitecture under the microscope. In comparison with the abovementioned cases, our case was performed in a hybrid operating room equipped with an ARTIS ICONO robotic C-arm cone beam computed topography (CBCT) scanner (Siemens Healthcare GmbH, Erlangen, Germany). The ICONO CBCT can provide intraoperative three-dimensional (3D) images that can be loaded instantly into the navigation system to help deal with any intraoperative event. Furthermore, intraoperative DSA in a hybrid operation room can better cope with an intraoperative emergency and provide confirmation of the total obliteration of AVMs.

AVM is a lesion within the brain parenchyma. Normally, the location of the nidus, depth, and drainage vein or feeding artery flow direction are determined by preoperative imaging and navigation. Augmented reality could directly inform the surgeon about the location, size, drainage vein or feeding artery structure and flow direction of the nidus, reducing the chance of disorientation, while decreasing the risk of intraoperative bleeding and providing a more intuitive surgical experience which can enhance the confidence of young staff during surgery. Moreover, disorientation caused by patient positioning can also be avoided with the use of AR technology.

In addition, the color discrepancy seen on the 3D projection makes it possible to see the spatial relationship between these feeding arteries, drainage veins and functional areas more clearly. Moreover, multiple feeding arteries can be coded with distinct colors in order to distinguish their depth, size and flow direction. With the processed image data loaded into navigation system, an operator can then perform microsurgical dissection and resection more safely under the AR image guidance. Even for operators who might be red–green colorblind, the system setup could adjust the discrepancy and contrast of colors.

When compared with the conventional navigation system, this microscope-based AR technique substantially lessens the problems that surgeons face when they are required to pay additional attention to the remote monitor while intraoperative navigation is needed. Through the use of the AR image system, all the navigation data can be transferred to the microscope and projected onto the surgical field. For neurosurgeons, fewer number of interruptions will result in both greater accuracy and safety.

However, certain limitations still exist when implementing the AR technique in microsurgical brain surgery. For example, it is still necessary for surgeons to remain aware that the projected object they see is not consistent with the exact anatomy, particularly during the stage after dura when the arachnoid membrane is opened, leading to the drainage of the cerebral spinal fluid (CSF). One consequence of the CSF drainage procedure is that certain cerebral parenchymal shrinkage may result in a deviation from intraoperative navigation. To resolve this problem, we performed an intraoperative CT and then integrated the instant image data into the navigation system to help assure accuracy of the AR image. Moreover, surgeons would be required to adapt to the display which is projected onto the surgical field under the microscope. With the surgical field covered by the hologram on the cortex, surgeons would be unable to perceive the precise depth of the surgical target [14,15]. Lastly, the costs surrounding set-up, maintenance and faculty training would be another limitation of this innovative technique.

## 6. Conclusions

In this case report, we have presented a patient diagnosed with right parietal lobe AVM who had undergone microsurgical resection surgery using the AR-assisted technique. This approach makes both the dissection and resection steps safer and more delicate to follow. As several innovations are occurring in AR technology today, it is likely that this novel technique will be increasingly adopted in both surgical applications and education. Although certain limitations exist, the procedure may still become more efficient and precise as this novel technology continues to develop further.

## Figures and Tables

**Figure 1 brainsci-13-00653-f001:**
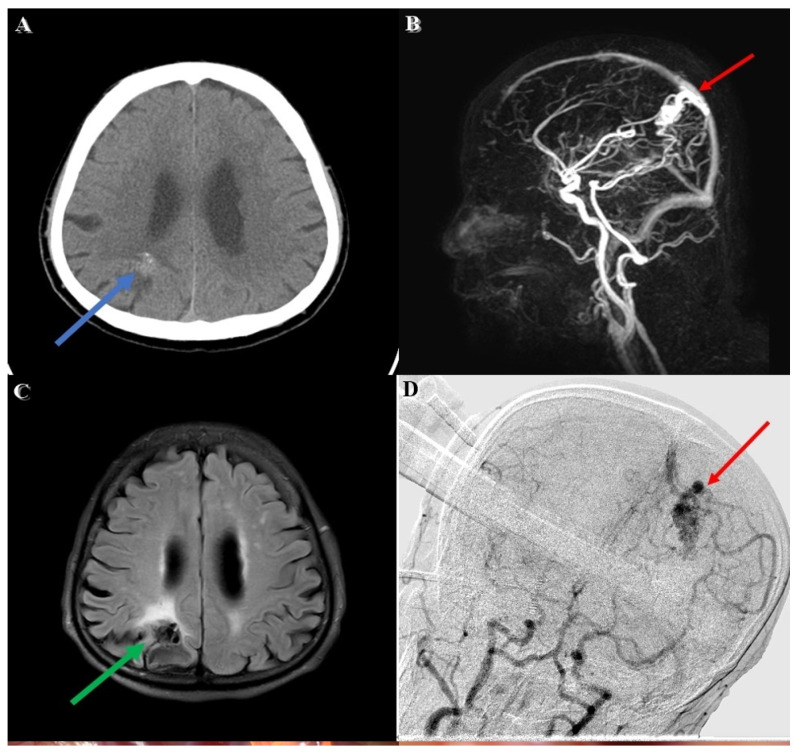
Preoperative and intraoperative images. (**A**) Right parietal suspected vascular lesion with hemorrhagic vestige from computed tomography (CT) (blue arrow). (**B**) Preoperative reconstructive magnetic resonance angiography (MRA) revealed the tangled vascular lesion (red arrow) (**C**) Flow voiding phenomenon was noted from T2-weighted image (green arrow). (**D**) Intraoperative digital subtraction angiography (DSA) for confirmation of lesion location.

**Figure 2 brainsci-13-00653-f002:**
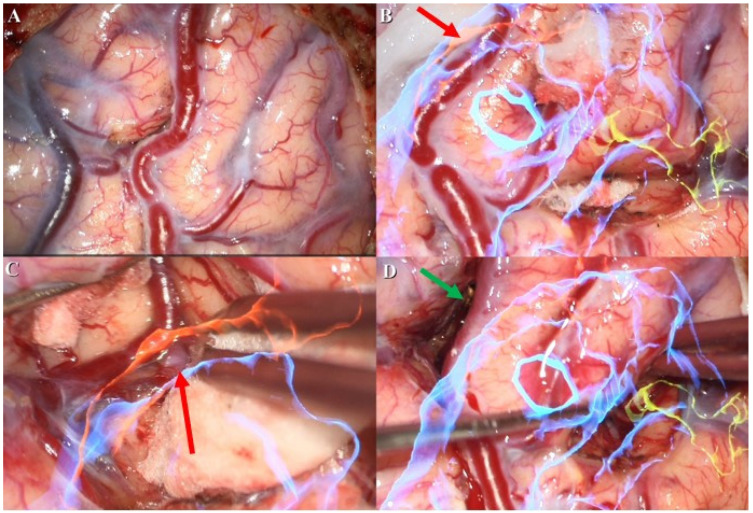
Microsurgery under assistance of AR technique. (**A**) Surgical field without AR hologram projection. (**B**) AVM nidus and feeding arteries were labeled with different color (blue: AVM nidus; red: feeding artery from middle cerebral artery; yellow: feeding artery from posterior cerebral artery); the red arrow also indicates feeding artery from middle cerebral artery. (**C**) Clipping and coagulation of the feeding artery. (**D**) Obliteration for another feeding artery from posterior cerebral artery; the green arrow indicates clip for feeding artery elimination.

## Data Availability

The data presented in this study are available upon request from the corresponding author. The data are not publicly available due to privacy restrictions.

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
