# Peer review of "Intraoperative Augmented Reality in Microsurgery for Intracranial Arteriovenous Malformation: A Case Report and Literature Review"

_brainsci, 2023, doi:10.3390/brainsci13040653_

Round 1

Reviewer 1 Report

This is a very good article describing AR assisted resection of AVM which is a evolving techhnology with promising feature.

This case report describes stereotactic microsurgical ressection of  AVM using augmented reality tecnology, this is a new technoligcal advanement in the field for microsurgical resection of AVM. This is a interesting paper describing authors experience however, AVM resection using augmented reality is previously been described. 

  In comparison with the existing literature this  case   was performed in a hybrid operating room equipped with an ARTIS ICONO robotic C-122 arm cone beam computed topography (CBCT) scanner, and intraopertive CT was performed and integrated in to AR images for better information and navigation. 

 References are appropraite 

 Study can be improved with incoporating comments of operating surgeon in this particular case. Was incorporating AR helped in idetifuing draing vein, Was AR useful in idetifying in complex angioarchtecture of AVM. 

Author Response

  1. Augmented reality could directly inform the surgeon about the location, size, drainage vein or feeding artery structure and flow direction of the nidus, reducing the chance of disorientation, while decreasing the risk of intraoperative bleeding and providing a more intuitive surgical experience which can enhance the confidence of young staff during surgery. Via the assistance of hologram projected on the cortical surface, the microsurgery would be more precise and safe avoiding abrupt bleeding.
  2. With the augmented reality technology got more frequently used in microsurgery, more incoporating comments from surgeons will be adopted to improve this novel technique.

Reviewer 2 Report

The authors decribe a novel technology by the Brainlab Company for the resection of brain arteriovenous malformations. The case is presented in a very nice and illustrative manner. 

The field of augmented reality supported operation has been elucidated by several publications since 2017. It would be interesting for the reader, how the surgical strategy for resection of arteriovenous malformations can be improved by using AR. Although the manuscript is well written, in my opinion it lacks some novelty.  

Author Response

  1. The AR technology could decrease the risk of intraoperative bleeding and providing a more intuitive surgical experience which can enhance the confidence of young staff during surgery. Via the assistance of the projected hologram on the surface, surgeons have more information and more precise location of the complex vessels. Not only the lesion location but also depth would be delicately provided.  

Reviewer 3 Report

In their revised paper, Li et al. reported their single-patient-based experience on the utility and feasibility of augmented reality (AR) in microsurgery for arteriovenous malformation (AVM) resection. The Authors found AR helpful in defining feeding vessels, minimizing the distraction of the surgeon. The paper is well-written, concise, and clear. The introduction provides sufficient background to the topic. The discussion remains properly written, providing some additional insight into the subject. References are correctly chosen. Conclusions reflect the data presented in the paper. I have found some minor shortcomings, which should be addressed before the paper's publication:

1) The manuscript carries some minor language errors. In my opinion, native-speaker English editing is not necessary. There are some minor flaws that I have noticed:

- Line 11: might result in – which might result in

- Line 62: was – were

- Line 83: with – in

- Line 97: proceed – proceeded

- Line 118: which – who

- Line 113: complete relief from previous symptoms – complete relief of previous symptoms

- Line 150: (…) there still exists certain limitations – (…) certain limitations still exist

2) Additionally, I have found some substantive flaws:

- Line 37: Since the Authors cited the paper on the Spetzler-Ponce classification [5], they should change the name of that grading system to: "Spetzler-Ponce classification" rather than "Spetzler-Martin grading system".

-  Line 38: Similarly, since the Authors referred to the Spetzler-Ponce classification, they should address Class A lesions rather than Grade I and Grade II (substantially the same lesions, while the terminology differs).

- Line 59: accidentally located – incidentally found? I do not think that AVM can be accidentally located. It can be diagnosed accidentally but not located.

- Line 60: computerized – computed

- Line 78: neuronavigational – neuronavigation

- Line 88: parietal-occipital craniotomy – parieto-occipital craniotomy

3) Finally, there are some concerns that I would like the Authors to address:

- Line 113: The Authors stated that “(…) complete relief from previous symptoms”. According to the data in the case description, the patient was initially asymptomatic (despite previous diagnostics due to headaches). Therefore, I doubt whether there were any symptoms that the patients could have recovered from. Please comment.

- Lines 113-115: I would appreciate additional figures of postoperative (follow-up) DSA, which confirmed the lesion excision.

- Lines 127-128: Did the Authors perform an intraoperative DSA for lesion confirmation? If yes, it would be of interest to add the Figure along with postoperative (follow-up) DSA.

Author Response

  1. The patient experienced a falling accident and was incidentally diagnosed with an intracranial AVM. The patient denied headaches or neurological symptoms prior to the accident. However, it is unclear if the patient experienced a seizure or loss of consciousness prior to the fall from a height of 3 meters. Following the accident, the patient experienced intermittent headaches for 6 months, which were eventually resolved through surgical intervention.
  2. For following-up of this patient, a post-operative MRA was selected as the imaging modality of choice. Confirmatory DSA had already been performed during the operation.
  3. We sincerely apologize for not being able to provide intraoperative DSA after the obliteration of the lesion. Although angiography was performed for confirmation, the image files were found to be corrupted. As a result, only the initial DSA taken prior to the craniotomy procedures could be preserved.  

Round 2

Reviewer 2 Report

I have no further comment.